# Amis *Pacilo* and Yami *Cipoho* are not the same as the Pacific breadfruit starch crop—Target enrichment phylogenomics of a long-misidentified *Artocarpus* species sheds light on the northward Austronesian migration from the Philippines to Taiwan

Chia-Rong Chuang[1], Chia-Lun Hsieh[1], Chi-Shan Chang[2], Chiu-Mei Wang[3], Danilo N. Tandang[1,4,5,6], Elliot M. Gardner[7,8], Lauren Audi[9], Nyree J. C. Zerega[10,11], Kuo-Fang Chung[1] *

**1** Research Museum and Herbarium (HAST), Biodiversity Research Center, Academia Sinica, Taipei, Taiwan, **2** National Museum of Prehistory, Taitung, Taiwan, **3** Department of Botany, National Museum of Natural Science, Taichung, Taiwan, **4** Philippine National Herbarium, National Museum of Natural History, National Museum of the Philippines, Manila, Philippines, **5** Biodiversity Program, Taiwan International Graduate Program, Academia Sinica, Taipei, Taiwan, **6** Department of Life Science, National Taiwan Normal University, Taipei, Taiwan, **7** International Center for Tropical Botany, Institute of Environment, Florida International University, Miami, Florida, United States of America, **8** National Tropical Botanical Garden, Kalaheo, Hawaii, United States of America, **9** Sackler Institute for Comparative Genomics, American Museum of Natural History, New York, New York, United States of America, **10** Negaunee Institute for Plant Conservation Science and Action, Chicago Botanic Garden, Glencoe, Illinois, United States of America, **11** Program in Plant Biology and Conservation, Northwestern University, Evanston, Illinois, United States of America

* bochung@gate.sinica.edu.tw

## Abstract

'Breadfruit' is a common tree species in Taiwan. In the indigenous Austronesian Amis culture of eastern Taiwan, 'breadfruit' is known as *Pacilo*, and its fruits are consumed as food. On Lanyu (Botel Tobago) where the indigenous Yami people live, 'breadfruit' is called *Cipoho* and used for constructing houses and plank-boats. Elsewhere in Taiwan, 'breadfruit' is also a common ornamental tree. As an essential component of traditional Yami culture, *Cipoho* has long been assumed to have been transported from the Batanes Island of the Philippines to Lanyu. As such, it represents a commensal species that potentially can be used to test the hypothesis of the northward Austronesian migration 'into' Taiwan. However, recent phylogenomic studies using target enrichment show that Taiwanese 'breadfruit' might not be the same as the Pacific breadfruit (*Artocarpus altilis*), which was domesticated in Oceania and widely cultivated throughout the tropics. To resolve persistent misidentification of this culturally and economically important tree species of Taiwan, we sampled 36 trees of Taiwanese *Artocarpus* and used the Moraceae probe set to enrich 529 nuclear genes. Along with 28 archived *Artocarpus* sequence datasets (representing a dozen taxa from all subgenera), phylogenomic analyses showed that all Taiwanese 'breadfruit' samples, together with a cultivated ornamental tree from Hawaii, form a fully supported clade

**Data Availability Statement:** Raw reads for all newly sequenced samples have been deposited in GenBank (Bioproject no. PRJNA855987).

**Funding:** This work was supported mainly by National Museum of Prehistory, Taiwan (N110011), and partly by Ministry of Science and Technology, Taiwan (MOST 109-2621-B-001-005-MY2) and Academia Sinica, Taiwan (AS-107-TP-B18) to Kuo-Fang Chung.

**Competing interests:** The authors have declared that no competing interests exist.

within the *A. treculianus* complex, which is composed only of endemic Philippine species. Morphologically, the Taiwanese 'breadfruit' matches the characters of *A. treculianus*. Within the Taiwanese samples of *A. treculianus*, Amis samples form a fully supported clade derived from within the paraphyletic grade composed of Yami samples, suggesting a Lanyu origin. Results of our target enrichment phylogenomics are consistent with the scenario that *Cipoho* was transported northward from the Philippines to Lanyu by Yami ancestors, though the possibility that *A. treculianus* is native to Lanyu cannot be ruled out completely.

## Introduction

Prior to western colonial expansion, Austronesian languages were the most widely spoken language family, covering 206˚ of longitude from Madagascar in the west to Rapa Nui (Easter Island) in the east and 72˚ of longitude from Taiwan in the north to Aotearoa (New Zealand) in the south [1]. Given the extraordinarily wide distribution, the quest into the ancestral homeland of Austronesian-speaking peoples (hereafter Austronesians) has been central in Pacific anthropology [1]. Recent studies in historical linguistics [1, 2], archaeology [3, 4], human genomics [5], and phylogeography of Pacific paper mulberry [6, 7] overwhelmingly support the Bellwood-Blust Model, positing that the epic Austronesian diaspora that eventually colonized almost all habitable Remote Oceanic islands by ca. 1300 A.D. started from the "Out of Taiwan" southward migration to the Philippines ca. 4200–4000 B.P. [3, 4, 8, 9]. However, Austronesian expansion and migration is by no mean unidirectional [10]. While all indigenous tribes of Taiwan's main island speak Formosan languages that constitute nine of the 10 subgroups of the Austronesian languages, the language of the Yami (also known as Tao) people of Lanyu (an offshore island of Taiwan also known as Botel Tobago; Fig 1) belongs to the Batanic languages, which are part of the Malayo-Polynesian languages (the 10th subgroup of the Austronesian language family) [1, 11]. As Yami are also culturally different from other Formosan indigenous peoples, it is generally assumed that the Yami are closely related to (and even descendants from) the Ivatan people of the Itbayat Island of the Batanes archipelago [12, 13], the northernmost Philippine islands ca. 100 km south of Taiwan and Lanyu [11, 14] across the Bashi Channel (Fig 1). The conjecture that Yami represents a northward Austronesian migration from the Philippines to Taiwan is also consistent with the idea that Neolithic Taiwan was part of the integral maritime trade networks around the South China Sea [15–17]. Surprisingly, however, archaeological and genetic studies both suggest that Yami are closer to Formosan indigenous peoples than the Ivatan [18, 19]. Bellwood and Dizon 2013 [20] surmised that the conflict among linguistic, archaeological, and human genetic evidence could have resulted from the replacement of earlier Formosan and non-Malayo-Polynesian by the Batanic speakers, suggesting a much more complicated origin of Yami people.

According to the "Farming-Language Dispersal Hypothesis", the key to Austronesians' rapid and successful expansion and migration was their early adoption of the farming and sedentary lifestyle that had enabled the increase and spread of Austronesian ancestral populations as well as the languages they spoke [4, 9, 22, 23]. Accompanying their migration, Austronesian ancestors also carried a suite of domesticated animals and crop species to increase the likelihood of surviving in remote and resource-poor islands [24]. Phylogeographic studies of these commensal organisms thus can serve as proxies for tracking Austronesian migration [25, 26], providing additional insights into the complexity of Polynesian seafaring [6, 27, 28]. Amongst

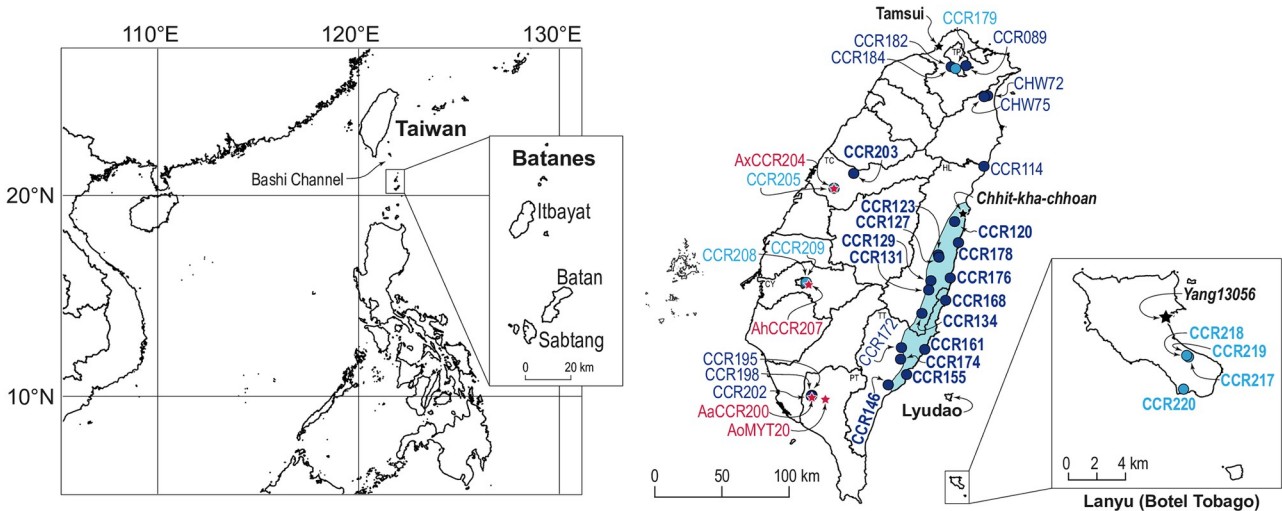

**Fig 1. Maps of Taiwan and the Philippines and sampling localities of Taiwanese *Artocarpus*.** In the sampling map, the traditional distribution range of Amis people [21] is highlighted in light blue. Dark blue circles denote sampling localities of 'breadfruit' (i.e., *Artocarpus treculianus*) originating from *Pacilo* (i.e., eastern Taiwan), while light blue circles denote sampling localities of 'breadfruit' originating from *Cipoho* (i.e., Lanyu). 'Breadfruit' samples collected from the traditional territories of Amis and Yami are also bold-faced. The samples and sampling localities of non 'breadfruit' *Artocarpus* are shown in red. See Table 1 for detailed sampling information. This work is licensed under a CC BY 4.0 license.

the ca. 200 plant species used by Yami people [29], 'breadfruit' has long been regarded as one of the possible plant species transported from the Philippines to Lanyu [14, 30].

Breadfruit [*Artocarpus altilis* (Parkinson) Fosberg] (Fig 2) is a fast-growing, low-maintenance, and high-yield traditional staple crop in Oceania [31, 32]. Though in the western world it is known primarily for its large, starchy, seedless fruits that were part of the infamous "Mutiny on the Bounty," in the Pacific, breadfruit comprises hundreds of cultivars (seeded and seedless), and it is a multipurpose tree species whose seeds, latex, inner bark, wood, leaves, and male inflorescences are traditionally used for food, medicines, house construction, boat making, carving, clothing, cordage, animal feed, wrapping, bird trapping, insect repellent, and fish poison [32, 33]. What we refer to as the 'fruit' of breadfruit throughout this manuscript is botanically a syncarp and is made up of tightly packed accessory fruits composed mainly of fleshy floral tissues. Breadfruit was domesticated in Near Oceania over 3,000 years ago and transported by Austronesian ancestors across Oceanic islands during their epic migration [24, 27]. Over millennia, hundreds of breadfruit cultivars (varying in fruiting season, fruit shape, color and texture of the flesh and skin, presence or absence of seeds, flavor, cooking and storage qualities, leaf shape, and horticultural needs) were selected by humans, and a small portion of that diversity was introduced to the Caribbean, Central and South America, Africa, India, Southeast Asia, and islands of the Indian Ocean [34]. Because of its high nutritional and medicinal values [35], breadfruit has been exalted as a sustainable crop for alleviating hunger and treating diabetes in Oceania and around the world [36, 37].

Despite its cultural and economic importance, species delimitation of breadfruit and its closest wild relatives has been difficult. This is in part due to its long taxonomic history and its complex domestication [38–41]. Numerous synonyms exist for the cultivated breadfruit. While the valid name is *Artocarpus altilis* (Parkinson) Fosberg [42], two synonyms are especially prevalent due to long history of use and taxonomic confusion: *A. communis* J.R.Forst. & G.Forst. [43] and *A. incisus* (Thunb.) L.f. [44]. Additionally, the circumscription of breadfruit has also varied. Jarrett 1959 [43] and Berg et al. 2006 [45], who mainly examined herbarium

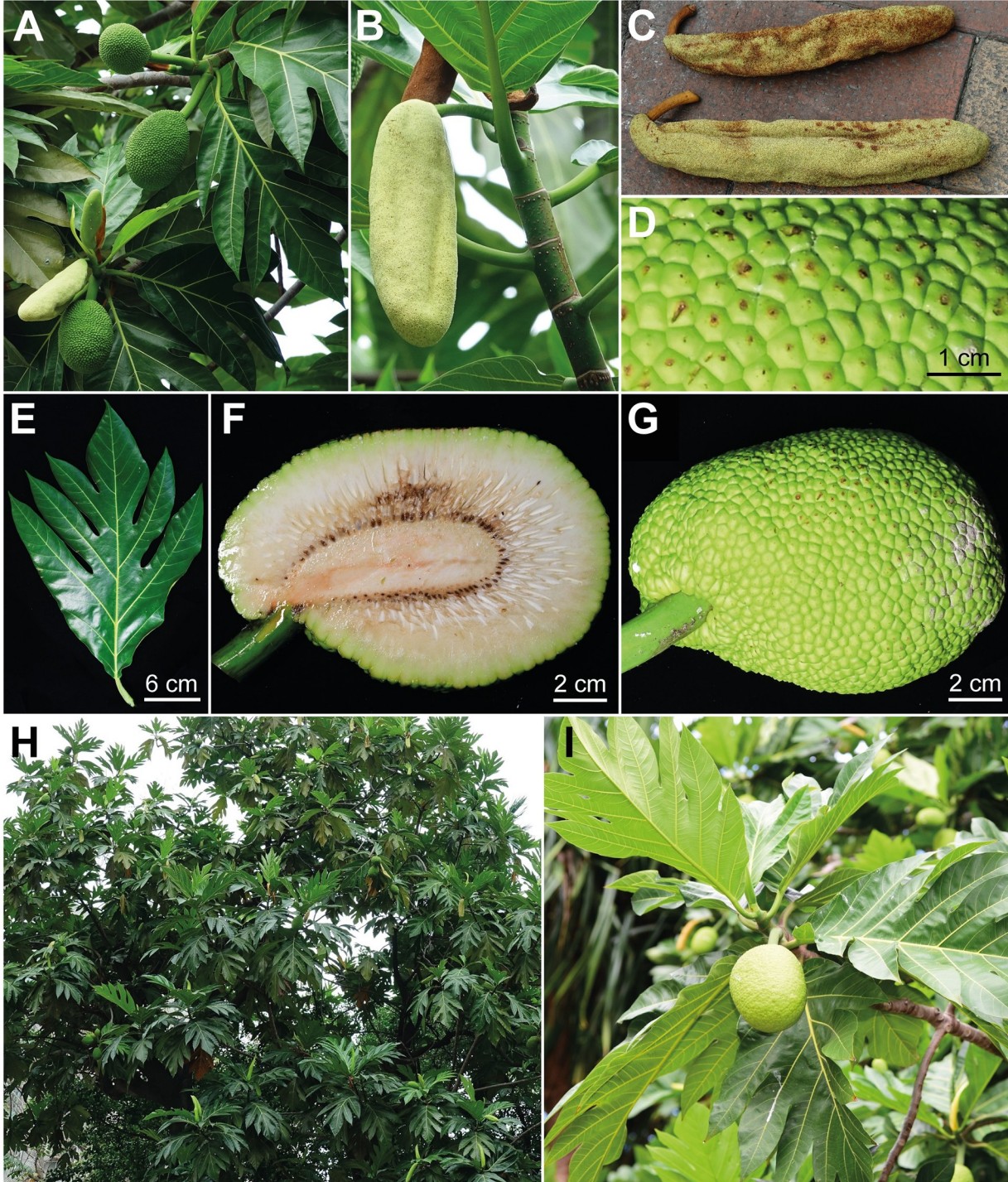

**Fig 2. *Artocarpus altilis* (Parkinson) Fosberg.** (A) Pistillate and staminate inflorescences. (B) Staminate inflorescence. (C) Fallen staminate inflorescences. (D) Syncarp surface. (E) Leaf. (F) Cross-section of syncarp. (G) Syncarp. (H) Tree (AaCCR200) planted on the campus of National Pingtung University (NPTU). (I) *Uru* planted on the campus of the University of Hawaiʻi at Mānoa. All photographs except for (I) was taken from tree planted in NPTU. A–C & H taken by C.-M. Wang, D–G by C.-L. Hsieh, and I by K.-F. Chung.

specimens, favored a broad approach that encompasses breadfruit's wild relatives. However, Zerega et al. 2005 [38], who worked with fresh specimens, molecular data, as well as herbarium specimens, took a narrower approach, recognizing *A. camansi* Blanco (breadnut) and *A. mariannensis* Trécul (*dugdug*) as distinct, closely related species of the cultivated breadfruit *A. altilis*. This latter approach has been widely adopted [32, 34, 45], including the current study.

In Taiwan, the local name for 'breadfruit' is "麵包樹 *miao-bao-shu*" (where "*mian-bao*" means bread and "shu" means tree in Mandarin Chinese), which is a common tree species occurring both in cultivation throughout much of the island and in forests of Lanyu [46]. The earliest traceable record of 'breadfruit' of Taiwan appeared in the diary (6[th] September 1890) of Rev. George Leslie Mackay (1844–1901) [47], a Canadian Presbyterian missionary who had profound influence on Taiwan's Christianity and medical education [48]. Mackay recorded "*Pat-chi-lút* Bread fruit tree" from *Chhit-kha-chhoan* [47] (Fig 1), a *Lam-si-hoan* (Amis) village in eastern Taiwan [49]. In his memoir "From Far Formosa", Mackay wrote that "Breadfruit (*Artocarpus incisa*)" was "*used by the aborigines exclusively*" [50]. In Amis' oral history, *Pacilo* (≡*Pat-chi-lút*) [51] (also known as '*Apalo* [51] / *Qapalo* [14], or *Facidol* called by central and southern Amis [14, 30]) was said to be introduced by Amis ancestors from overseas [52]. Nowadays, *Pacilo* is still commonly planted in eastern Taiwan especially around Amis households as food [51] and has become a popular ethnic cuisine of eastern Taiwan.

In Yami language, 'breadfruit' is called *Cipoho* or *Ciporo*, almost identical to *Cipoho* or *Tipoho* of the other Batanic languages [14]. In forests managed by Yami people, *Cipoho* is the third most dominant tree species [53]; traditionally, however, fruits of *Cipoho* are rarely consumed as food on Lanyu [54, 55]. Instead, Yami people use the light red wood of *Cipoho* for making houses, hats, plates, mortars (for crushing millet seeds), and the plank-boats *Tatala* and *Cinedkeran* that are central to the Yami culture [55–59]. Given the phonetic similarity of the names for 'breadfruit' in Batanic languages [14] and its appearance as a timber tree in managed and semi-naturalized forests of Lanyu [53, 60, 61], *Cipoho* has long been regarded as non-native [30, 61], possibly introduced by Yami ancestors along with their northward migration from Batanes Island [30].

However, in the earliest record of Lanyu's 'breadfruit', which was documented by Yasusada Tashiro [60] in 1900 A.D. during the early period of the Japanese occupation, the species appeared to have been confused with the jackfruit "*Artocarpus integrifolia* L." (= *A. heterophyllus* Lam.), which was introduced during the Dutch colonization (1624–1662 A.D.) and has been widely grown in Taiwan since [52, 62]. In 1900 A.D., Tashiro was amongst the earliest botanists who ever visited Botel Tobago, and in "A guide to planting trees in urban Taiwan," he recorded 'breadfruit' as growing naturally and abundantly on Botel Tobago and commonly planted around indigenous villages of eastern Taiwan [60]. To promote 'breadfruit' as an urban tree species suitable even in northern Taiwan, Tashiro also referred to a 'breadfruit' tree grown in Mackay's residence in Tamsui (Fig 1) where he lived during his mission. Based on a widely circulated anecdote, that vibrant tree was grown from a *Pat-chi-lút* seedling given to Mackay by the *Nan-shih* Amis [63]. Likely due to Tashiro's promotion [60, 64], 'breadfruit' is now widely cultivated throughout Taiwan and is amongst the most common tree species planted on school campuses [65–67]. Because 'breadfruit' trees are easily accessible, its natural compounds and pharmaceutical properties have been extensively studied in Taiwan [68–70].

Although 'breadfruit' is fairly common in Taiwan, its taxonomy has been confusing. In addition to early confusion with the jackfruit [60, 71], Li 1963 [72] also confused the species with *A. lanceolatus* Trécul (= *A. lamellosus* Blanco) in the "Woody Flora of Taiwan," an important post World War II floristic work. Moreover, the above mentioned three common binomials used for the true Pacific breadfruit, (i.e., *Artocarpus altilis* [53, 69, 73, 74], *A. communis* [52, 53, 75–80], and *A. incisus* [29, 46, 61, 64, 81–84]) have been variously adopted even within a

single publication [53], resulting in much confusion (S1 Appendix). And yet, the taxonomy of Taiwanese 'breadfruit' has never been rigorously studied. In recent phylogenomic studies of *Artocarpus* using target enrichment (Hyb-Seq), an herbarium sample collected from Lanyu (*Yang13056*), initially identified as '*Artocarpus incisus*', was determined to be *Artocarpus treculianus* Elmer based on morphology and grouped with *NZ203* (a specimen of *A. treculianus* collected from cultivation in Hawaii). Both specimens were referred to as "*A. treculianus* 'ovatifolius*" for their morphological similarities with the type specimen of *A. ovatifolius* Merr. [85]. These samples were within the 'Philippinensi' clade of Sect. *Artocarpus*, Subgenus *Artocarpus*, a clade comprising species thought to be endemic to the Philippines [86]. Since *A. treculianus* (including *A. ovatifolius* Merr.) has long been considered endemic to the Philippines [43, 45, 86], the likely persistent misidentification of the Taiwanese 'breadfruit' prompted us to conduct the present study. Additionally, considering the close geographic proximity and phytogeographic similarity between Lanyu and the Philippines [87] (Fig 1), the native status of *A. treculianus* has to be reconsidered. Indeed, in the earliest record by Tashiro [60], *Cipoho* was regarded as a native species on Lanyu, though latter scholars assumed that the species was not native since it was the "breadfruit" [30, 61]. By taking advantage of recent phylogenomic studies in *Artocarpus* [88] using Hyb-Seq [85, 86, 89], we asked the following questions: (1) How many species of *Artocarpus* constitute the 'breadfruit' species of Taiwan and what is the correct name of the most common 'breadfruit' of Taiwan? (2) Is *Cipoho* native to Lanyu? and (3) What are the likely geographic origins of these 'breadfruit' species? By answering these questions, we also hope to contribute to our understanding of Yami's ancestry.

## Materials and methods

### Sampling

In Taiwan, the "breadfruit" is considered to be *Artocarpus incisus* (Thunb.) L.f. (= *A. altilis*), which in "The Red List of Vascular Plants of Taiwan 2017" is listed as "Least Concern" [84]. So only oral permissions from local land owners were needed to sample the species. Since there is a strong indigenous consciousness on Lanyu, we also obtained a collecting permit from the Lanyu Township Office. Prior to sampling, we thoroughly examined specimens in major herbaria (TAI, TAIF, TNM, and HAST [90]), literature [46, 72, 74, 76, 79, 81], and online information pertaining to the 'breadfruit' of Taiwan. Our preliminary investigation concluded that the morphology of almost all Taiwanese 'breadfruit' trees matched *A. treculianus* as recently circumscribed by Gardner and Zerega 2021 [86], rather than the true Pacific breadfruit, *A. altilis*. Because *A. treculianus* is considered a species complex and its taxonomy remains unsettled [86], we proceeded to sample 'breadfruit' from multiple localities of Taiwan, with a special focus on the traditional territory of the Amis of eastern Taiwan and Lanyu (Fig 1). In addition to the Taiwanese 'breadfruit', a genuine *A. altilis*, which is extremely rare in Taiwan, was located in National Pingtung University (NPTU) and sampled (Fig 2). We also sampled one jackfruit (*A. heterophyllus*), one *A. odoratissimus* Blanco [91], and one *A. xanthocarpus* from National Chiayi University (NCYU), National Pingtung University of Science and Technology (NPUST), and National Museum of Natural Science (NMNS), respectively, to improve *Artocarpus* sampling within Taiwan. Voucher information of the 36 samples collected in current study is detailed in Table 1. Voucher specimens were deposited in the Herbarium of Academia Sinica, Taiwan (HAST). To assure species identification of our samples, 28 already-sequenced accessions (S1 Table), including 10 accessions of the 'Philippinensi' clade *sensu* Gardner and Zerega 2021 [86], two accessions of *A. xanthocarpus*, one accession of *A. heterophyllus*, two accessions of *A. mutabilis* Becc. (*Pingan*), one accession of *A. odoratissimus* (*Lumok*) [91], two accessions of *A. altilis*, and five accessions of *A. camansi*, were also included, with *A. frutescens*

**Table 1. Sampling information of *Artocarpus* species of Taiwan.**

| Code | Species | Voucher | HAST specimen ID | Locality | Coordinate |
|---|---|---|---|---|---|
| AaCCR200 | *A. altilis* | *C.-R. Chuang 200* | 145709 | Pingtung, NPTU[a] | 22.6650, 120.5043 |
| AhCCR207 | *A. heterophyllus* | *C.-R. Chuang 207* | 145712 | Chiayi, NCYU[b] | 23.4688, 120.4859 |
| AoMYT20 | *A. odoratissimus* | *M.-Y. Tsai 20* | 145690 | Pingtung, NPUST[c] | 22.6454, 120.6057 |
| CHW72 | *A. treculianus* | *H.-W. Chen 72* | 145811 | Yilan, Jiaoxi | 24.8236, 121.7707 |
| CHW75 | *A. treculianus* | *H.-W. Chen 75* | 145812 | Yilan, Jiaoxi | 24.8164, 121.7458 |
| CCR089 | *A. treculianus* | *C.-R. Chuang 089* | 145784 | Taipei, AS[d] | 25.0399, 121.6154 |
| CCR114 | *A. treculianus* | *C.-R. Chuang 114* | 145644 | Hualien, Sioulin | 24.3166, 121.7442 |
| CCR120 | *A. treculianus* | *C.-R. Chuang 120* | 145653 | Hualien, Shoufeng | 23.9196, 121.5335 |
| CCR123 | *A. treculianus* | *C.-R. Chuang 123* | 145658 | Hualien, Guangfu | 23.6812, 121.4166 |
| CCR127 | *A. treculianus* | *C.-R. Chuang 127* | 145785 | Hualien, Guangfu | 23.6638, 121.4208 |
| CCR129 | *A. treculianus* | *C.-R. Chuang 129* | 145786 | Hualien, Rueisuei | 23.4951, 121.3633 |
| CCR131 | *A. treculianus* | *C.-R. Chuang 131* | 145787 | Hualien, Yuli | 23.4281, 121.3477 |
| CCR134 | *A. treculianus* | *C.-R. Chuang 134* | 145788 | Hualien, Fuli | 23.2600, 121.2989 |
| CCR146 | *A. treculianus* | *C.-R. Chuang 146* | 145789 | Taitung, Taitung | 22.7469, 121.0569 |
| CCR155 | *A. treculianus* | *C.-R. Chuang 155* | 145790 | Taitung, Beinan | 22.8212, 121.1890 |
| CCR161 | *A. treculianus* | *C.-R. Chuang 161* | 145791 | Taitung, Chenggong | 23.0014, 121.3194 |
| CCR168 | *A. treculianus* | *C.-R. Chuang 168* | 145792 | Taitung, Changbin | 23.3541, 121.4692 |
| CCR172 | *A. treculianus* | *C.-R. Chuang 172* | 145793 | Taitung, Guanshan | 23.0143, 121.1517 |
| CCR174 | *A. treculianus* | *C.-R. Chuang 174* | 145794 | Taitung, Luyeh | 22.9313, 121.1455 |
| CCR176 | *A. treculianus* | *C.-R. Chuang 176* | 145795 | Hualien, Fengbin | 23.5147, 121.5024 |
| CCR178 | *A. treculianus* | *C.-R. Chuang 178* | 145796 | Hualien, Shoufeng | 23.7685, 121.5611 |
| CCR179 | *A. treculianus* | *C.-R. Chuang 179* | 145797 | Taipei, NTU[e] | 25.0186, 121.5407 |
| CCR182 | *A. treculianus* | *C.-R. Chuang 182* | 145798 | Taipei, TPBG[f] | 25.0317, 121.5091 |
| CCR184 | *A. treculianus* | *C.-R. Chuang 184* | 145799 | Taipei, TPBG[f] | 25.0318, 121.5079 |
| CCR195 | *A. treculianus* | *C.-R. Chuang 195* | 145800 | Pingtung, Pingtung | 22.6726, 120.5088 |
| CCR198 | *A. treculianus* | *C.-R. Chuang 198* | 145801 | Pingtung, Pingtung | 22.6588, 120.5088 |
| CCR202 | *A. treculianus* | *C.-R. Chuang 202* | 145802 | Pingtung, NPTU[a] | 22.6666, 120.5048 |
| CCR203 | *A. treculianus* | *C.-R. Chuang 203* | 145803 | Taichung, Shigang | 24.2652, 120.8093 |
| CCR205 | *A. treculianus* | *C.-R. Chuang 205* | 145804 | Taichung, NMNS[g] | 24.1586, 120.6685 |
| CCR208 | *A. treculianus* | *C.-R. Chuang 208* | 145805 | Chiayi, CYBG[h] | 23.4833, 120.4687 |
| CCR209 | *A. treculianus* | *C.-R. Chuang 209* | 145806 | Chiayi, CYBG[h] | 23.4833, 120.4672 |
| CCR217 | *A. treculianus* | *C.-R. Chuang 217* | 145807 | Taitung, Lanyu | 22.0281, 121.5780 |
| CCR218 | *A. treculianus* | *C.-R. Chuang 218* | 145808 | Taitung, Lanyu | 22.0291, 121.5767 |
| CCR219 | *A. treculianus* | *C.-R. Chuang 219* | 145809 | Taitung, Lanyu | 22.0291, 121.5767 |
| CCR220 | *A. treculianus* | *C.-R. Chuang 220* | 145810 | Taitung, Lanyu | 22.0061, 121.5745 |
| AxCCR204 | *A. xanthocarpus* | *C.-R. Chuang 204* | 145711 | Taichung, NMNS[g] | 24.1590, 120.6676 |

[a]NPTU: National Pingtung University.

[b]NCYU: National Chiayi University.

[c]NPUST: National Pingtung University of Science and Technology.

[d]AS: Academia Sinica.

[e]NTU: National Taiwan University.

[f]TPBG: Taipei Botanical Garden.

[g]NMNS: National Museum of Natural Science.

[h]CYBG: Chiayi Botanical Garden.

Renner, *A. limpato* Miq., and *A. papuanus* Renner selected as outgroups according to Gardner et al. 2021 [85].

## DNA extraction, library preparation, and sequencing

Genomic DNA was extracted from 0.01 g silica-dried young leaves using the modified CTAB method [92], purified using NEB (Ipswish MA, USA) Monarch® PCR & DNA Cleanup Kit (5 μg), and qualified and quantified using 1% agarose gel and a Qubit™ Fluorometer 4 (Invitrogen, Life Technology, CA, USA). DNA was randomly fragmented using Bioruptor® Pico B01060010 (Diagenode, Liège, Belgium) into 300–400 bp fragments, and libraries were prepared using NEBNext® Ultra™ II DNA Library Prep Kit for Illumina® (NEB, Ipswish MA, USA). Libraries were then quantified using Qubit™ Fluorometer 4 and Fragment analyzer 5200 (Agilent, CA, USA) to ensure the average final insert sizes of 450–550 bp. Three pools of 12 libraries were enriched for 529 targeted phylogenetic markers designed for *Artocarpus*-Moraceae [89] using MYbaits v5.01 custom probes (MYcroarray, Ann Arbor, MI, USA) following manufacturer's protocol, and subsequently reamplified with 14 PCR cycles based on parameters detailed in Gardner et al. 2016 [89]. A total of 36 enriched samples of *Artocarpus* were pooled with 22 samples of other projects into a lane and sequenced by NGS High Throughput Genomics Core at Biodiversity Research Center, Academia Sinica (BRCAS) using an Illumina HiSeq 2500 (Illumina, San Diago, CA, USA) with paired end 150 bp HiSeq Rapid mode. Raw reads for all newly sequenced samples have been deposited in GenBank (Bioproject no. PRJNA855987).

## Sequence assembly and phylogenetic analysis

The sequencing quality of raw reads of 36 newly sequenced and 28 archived samples were evaluated using FastQC v0.11.5 [93] and trimmed using Trimmomatic v0.39 [94] to filter out low-quality reads under quality score 20 with 4-bp window size, discarding any reads under 50 bp. Following the procedures detailed in Gardner et al. 2021 [85], the trimmed paired-reads were assembled using HybPiper 1.3.1 [95], a pipeline that produces gene-by-gene, reference-guided assemblies for phylogenomic analyses. In brief, trimmed paired-reads were first mapped to reference genes [85] using BWA v0.7.17 [96] and those successfully mapped were assembled into contigs using SPAdes v3.13.0 [97]. Exonerate [98] was then used to align the assembled contigs to their associated target sequences that contains exons and introns. To evaluate impact of noncoding sequences on our analyses, we used HybPiper to generate both the default 'exon' output as well as 'supercontig' sequences (consisting of exons plus flanking noncoding sequences). In the 'exon' dataset, genes that received a 'paralogs warning' from HybPiper were all removed. Assembled sequences that were shorter than 100 bp or 20% of average length as well as genes possessed by less 30 samples were removed, subsequently aligned with MAFFT v7.310 [99], and then trimmed by trimAl 1.2 rev59 [100] to filter out columns containing over 75% gaps. For the 'supercontig' alignment, an additional pipeline Putative Paralogs Detection (PPD) was used for potential paralogs detection and removal using a default setting [101], subsequently aligned by MAFFT v7.310 [99], and then trimmed by trimAl 1.2 rev59 [100] to filter out the column containing over 49% gaps. Gene trees were reconstructed by IQ-TREE 2.0 [102] using the `paramter-m MFP-B 1000`. The coalescent based species trees were then calculated by ASTRAL v5.7.8 using default settings [103].

## Results and discussion

A total of 30,843 Mb of raw reads were acquired for the 36 newly sequenced samples, with an average of 857 Mb per sample. For the 28 archived samples, a total of ca. 6,804 Mb of raw reads

were downloaded, with an average of ca. 243 Mb per sample. Statistics of the assembled quality and summaries of the exon and supercontig alignments are detailed in S1 Table. On average, 524 loci were assembled for each sample, with the final species trees including 296 loci and 524 loci in the 'exon' and 'supercontig' datasets, respectively. The exon alignment contained 357,666 characters, while the supercontig alignment contained 890,946 bp.

*Artocarpus* is a taxonomically challenging genus [38, 86, 88, 104]. Recent studies using target enrichment phylogenomics suggest that the morphology-based taxonomy has both under- and overestimated species diversity in *Artocarpus*. For instance, Gardner et al. 2022 [91] demonstrated that *A. odoratissimus s.l.* circumscribed in the Linnaean taxonomy for two centuries [45] actually comprises two distinct species *Lumok* (= *A. odoratissimus s.s.*) and *Pingan* (= *A. mutabilis* Becc.) long recognized by indigenous populations in Borneo [91]. Meanwhile, *A. multifidus* F.M.Jarrett, a species recognized by its number of leaf lobes, is genetically indistinguishable from *A. pinnatisectus* Merr. [86].

In ASTRAL trees reconstructed based on both exons and the supercontigs (Fig 3), phylogenetic relationships of all major clades are fully supported and identical to those proposed in Gardner and Zerega 2021 [86] and Gardner et al. 2021 [85]. Specifically, *Artocarpus* subg. *Prainea* is sister to the clade of Subg. *Pseudojaca* + Subg. *Artocarpus*. Within Subg. *Artocarpus*, Sect. *Cauliflori* (*sensu* Gardner and Zerega 2021 [86]) is sister to the clade of Sect. *Duricarpus* + Sect. *Artocarpus*. Within Sect. *Artocarpus*, Clade 'Incisifolii' is sister to Clade 'Philippinensi' (the 'Rugosi' clade not having been sampled for this study). Within the Clade 'Incisifolii', our sampled *A. altilis* (AaCCR200) cultivated in Taiwan is placed sister to a clade composed of two Pacific *A. altilis* samples (K7 from Somoa and V8 from French Polynesia), confirming our species identification. Together with two *A. treculianus* (*NZ203* and *Yang13056*) sequenced by Gardner et al. 2021 [85], all of our 32 samples of the Taiwanese 'breadfruit' form a fully supported clade (Taiwan 'breadfruit' clade) within the *A. treculianus* complex clade, which is sister to the clade containing *A. nigrescens* and *A. pinnatisectus* that are both endemic to the Philippines.

The placement of the 32 samples of the Taiwanese 'breadfruit' within the *Artocarpus treculianus* complex confirms our conjecture that this common plant has been misidentified since Rev. Mackay's earliest documentation [50]. Our ASTRAL trees also confirm the identification of *A. altilis*, *A. heterophyllus*, *A. odoratissimus*, and *A. xanthocarpus* (Fig 3) in Taiwan. Specifically, within Sect. *Duricarpus*/Clade 'Asperifolii', AoMYT20 and NZ618 (*A. odoratissimus s.s.* "*Lumok*") form a clade sister to the clade of *A. mutabilis* "*Pingan*", indicating that the Taiwanese *A. odoratissimus s.l.* is the *Lumok* [91].

Taxonomy of the *A. treculianus* complex remains unsettled [86], as there is considerable variation in leaf shape and pubescence, as well as the length of staminate inflorescence (ranging from 1–21 cm), and in the shape of the syncarp. Nevertheless, morphology of the Taiwanese 'breadfruit' sampled in the current study (Fig 4) differs from the closely allied *A. blancoi* and matches *A. treculianus* as circumscribed by Gardner and Zerega 2021 [86], confirming our preliminary identification. It is important to note that our current understanding of *A. treculianus* has been based on limited herbarium collections, which often do not well represent the morphological variation, even within an individual tree. For instance, leaves of *A. treculianus* observed in Taiwan often show considerable variation during different developmental stages (Fig 4L) as well as within a single tree (Fig 4M). An even greater variation is observed in staminate inflorescences (Fig 5), which can be important in the classification of *Artocarpus* species [86].

Since Jarrett's [43] seminal taxonomic work, *Artocarpus treculianus* has been regarded as an endemic taxon of the Philippines, with its northern distributional limit in the Batanes Islands [45, 86] where this "vulnerable" species grows both around human habitations as well

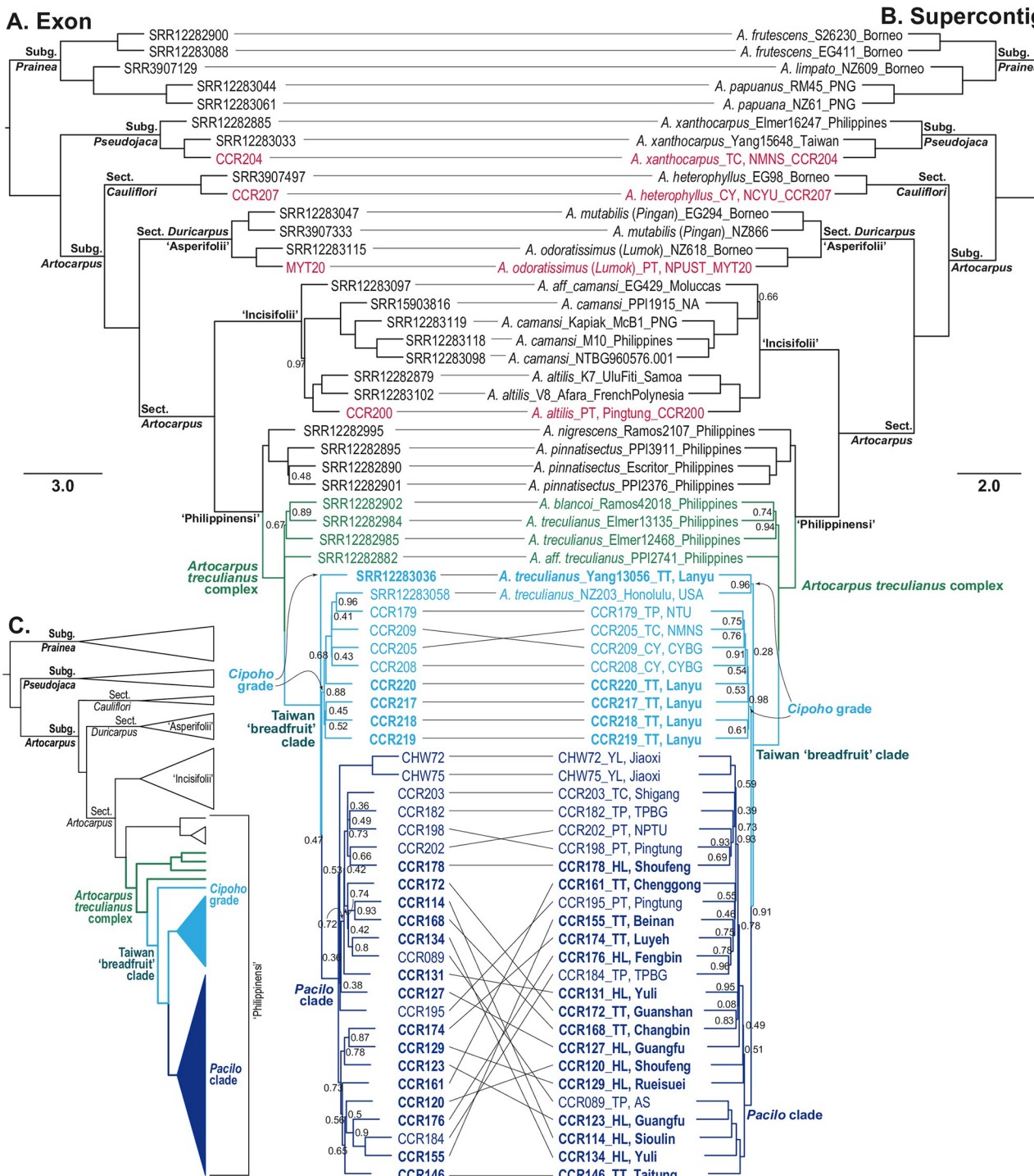

**Fig 3. Comparison between ASTRAL trees reconstructed based on exon (A) and supercontig (B) datasets.** Samples collected in present study and *A. treculianus* complex are shown in colors, with samples of non 'breadfruit' *Artocarpus* in red, Philippine *A. treculianus* in green, *Cipoho* grade in light blue, and *Pacilo* clade in dark blue. Sample names of 'breadfruit' collected from the traditional territory of Amis and Lanyu are bold-faced. Node labels denote posterior probabilities that are not 1.00. A simplified cladogram is shown in **C**. CY: Chiayi; HL: Hualien; PT: Pingtung; TC: Taichung; TP: Taipei; TT: Taitung. See Table 1 for other abbreviations.

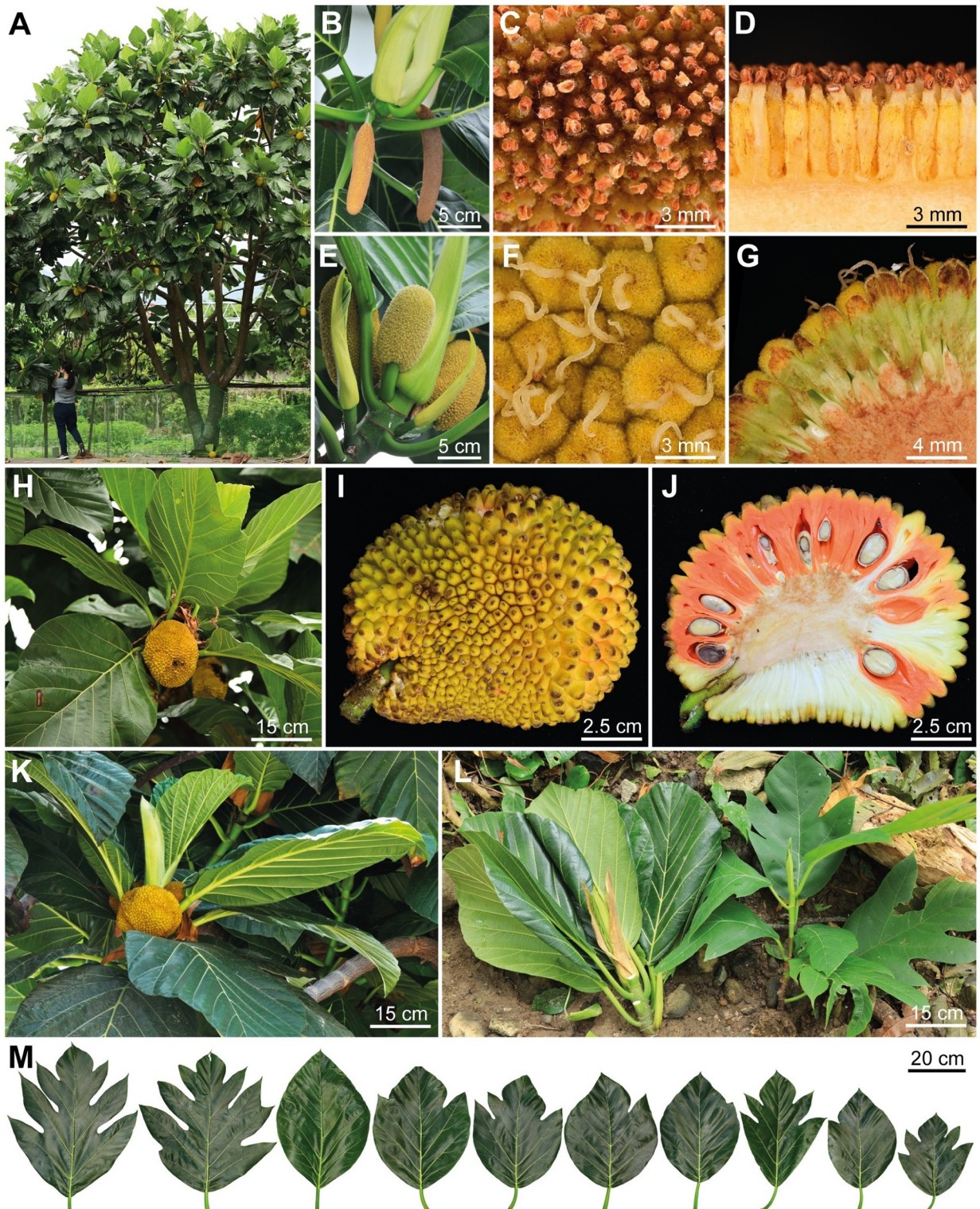

**Fig 4.** ***Artocarpus treculianus* Elmer.** (A) Tree. (B) Staminate inflorescences. (C) Staminate flowers, top view. (D) Staminate flowers, side view. (E). Pistillate inflorescences and stipules. (E) Pistillate flowers, top view. (G) Pistillate flowers, side view. (H) Syncarps and leaves. (I) Syncarp surface. (J) Cross-section of syncarp. (K) Syncarps and leaves of a tree from Batan Island. (L). A branch from a mature tree (left) and a seedling (right). (M) Variation of leaf shapes. Except for (K), all other photographs were taken in Taiwan. A, B, E, H, & L taken by K.-F. Chung, C, D, F, G, I, J, & M by C.-L. Hsieh, and K by D. N. Tandang.

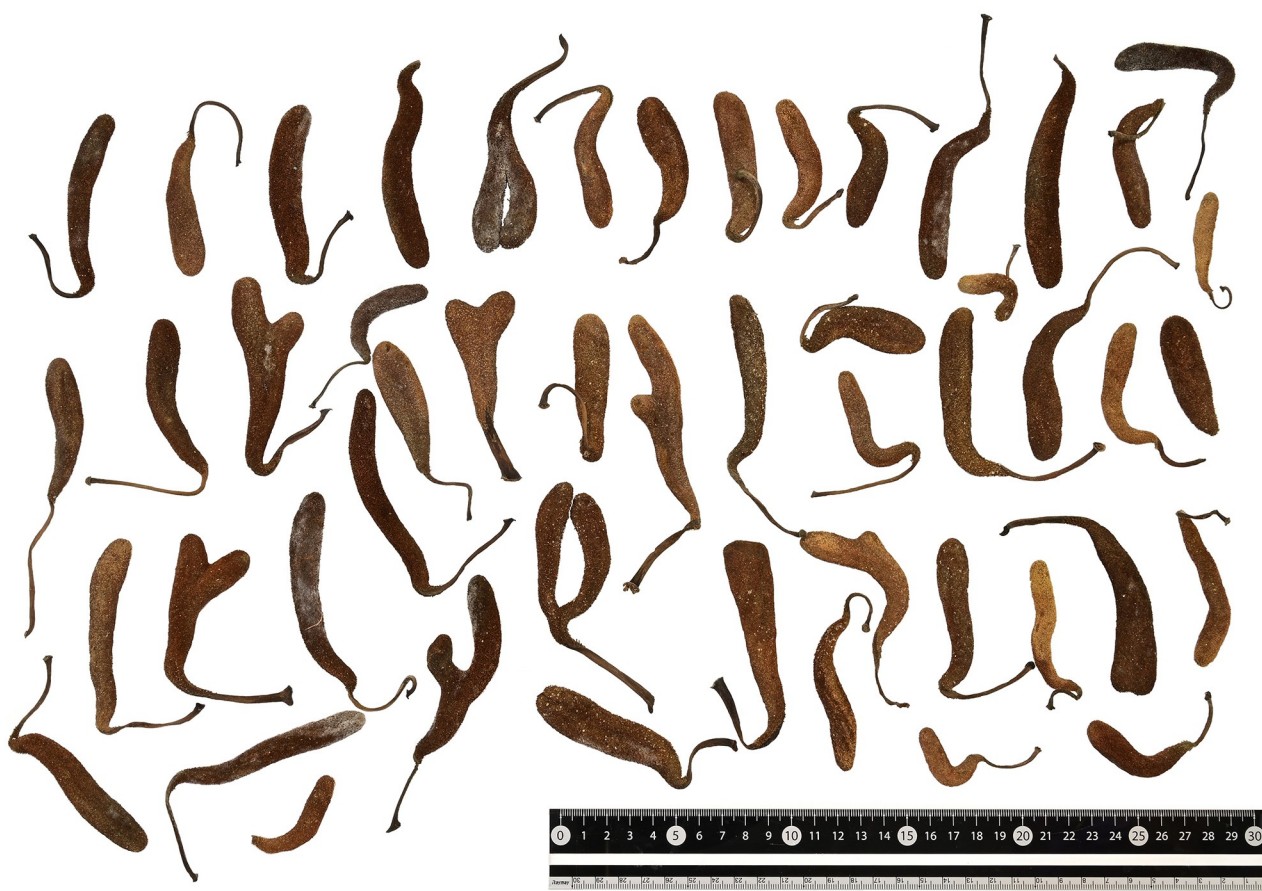

**Fig 5. Variation in staminate inflorescences in *Artocarpus treculianus*.** These staminate inflorescences all fell from a single tree planted on the campus of Academia Sinica, Taipei (CCR089) and gathered in June 1, 2022. Taken by K.-F. Chung.

as "in secondary forests and edges of forests" [105]. As a popular shade tree, its fruits are also eaten and leaves of this tree (Fig 4K) are also eaten and used as plates and to wrap food during festivities [105]. Additionally, its wood is used for making *Bangka* (a type of fishing boat common in the Philippine coastal villages), *Bangka* trusses, and doors and windows of traditional houses. Moreover, its white latex is cooked and used as bird-trap [105]. All these traditional utilities are highly similar to the traditional Yami culture [29, 55, 59, 75]. The presence of *A. treculianus* in Lanyu thus is consistent with the scenario of a northward transport by ancestors of speakers of the Batanic languages, providing a commensal species likely brought by the 'into Taiwan' Austronesian migration [14, 30]. On the other hand, the possibility that *A. treculianus* is a native species of Lanyu cannot be ruled out, given the close geographical proximity and phytogeographic similarity between Lanyu and the Philippines [87]. Indeed, Tashiro recorded the tree was native and abundant in the forests of Lanyu after his first visit to the island in 1900 A.D. [60]. Though seeds and large syncarps of *Artocarpus* (Figs 2 and 4) are generally assumed to be dispersed by forest ungulates [45], Gardner et al. 2022 [106] proposed a 1,600 km bird-mediated dispersal across Wallacea for the origin of *A. buyangensis* E.M.Gardner, Jimbo & Zerega, a recently described species endemic to Manus Island, Papua New Guinea. Given this, a seed of *A. treculianus* carried by a bird or possibly a flying fox (*Pteropus sp.*) flying across the

185 km-wide Bashi Channel (Fig 1) separating Lanyu and the Batanes Island is highly plausible. To test these alternative scenarios, further sampling in the Batanes Islands and from throughout the Philippines is essential.

Despite limited sampling, our ASTRAL trees reveal a marked phylogeographic structure. Considering just the samples collected from traditional territories of Amis and Yami (bold-faced in Fig 3), the Amis samples form a fully supported clade (i.e., *Pacilo* clade) deriving from a paraphyletic grade consisting of the Lanyu samples (i.e., *Cipoho* grade), suggesting that the Amis samples originated from Lanyu. This inference is consistent with the Amis tale that their ancestors came from the overseas islands Sanasai (i.e., Lyudao, an island. 73 km north of Lanyu; Fig 1) and/or Vutul (i.e., Botel Tobago) [107, 108]. On the contrary, the lack of structure within the *Pacilo* clade suggests that this popular fruit crop of the Amis people may have been extensively translocated across eastern Taiwan. Additionally, within the *Pacilo* clade, the generally poor support values and highly discordant and entangled relationships between the exon and supercontig trees indicate the reticulated nature of tokogenetic relationships. Nevertheless, based on the ASTRAL trees (Fig 3), it is evident that *A. treculianus* cultivated in CYBG (CCR208 and 209), NMNS (CCR205), and NTU (CCR179) originated from Lanyu, while trees planted in AS (CCR089), TPBG (CCR182 and 184), NPTU (CCR202), and other localities (CCR195, 198, and 203, and CHW72 and 75) were derived from eastern Taiwan (Fig 1), indicating multiple introductions of the 'breadfruit' to various parts of the island. Additionally, the placement of *NZ203*, a sample cultivated at the Fort de Russey Park in Honolulu, suggest a possible Lanyu origin. Our data thus demonstrate that the *Artocarpus*-Moraceae probe set [89] is efficient in detecting phylogeographic structure even on a small geographic scale, providing a genomic tool for unraveling the origin of *A. treculianus* with a more comprehensive sampling.

The correction of a persistent misidentification of the Taiwanese 'breadfruit' also underscores the possibility that the taxonomy of many common species might still be problematic. Unlike rare species, common plants may be less attractive for taxonomic research because their taxonomy is usually just assumed to be well-studied. On the other hand, because common species are often easily accessible, they are often used for applied research. For instance, a quick 'Google Scholar' search with the key words 'breadfruit' and 'Taiwan' resulted in more than 30 research articles of natural compound and pharmaceutical studies of either '*Artocarpus altilis*' or '*A. communis*', which were all mis-identifications of *A. treculianus* (S1 Appendix). This has major impacts on downstream use of such data! Although *A. treculianus* and *A. altilis* are both members of Sect. *Artocarpus*, the crown age of the diversification of the two species was estimated at ca. 25 million years ago in the Oligocene [109]. The long time elapsed since the split of their most recent common ancestor surely would have resulted in numerous evolutionary changes including the properties of chemical compounds. If readers of those research articles are not aware of this species mis-identification, the scientific values of these studies would be greatly undermined [110].

## Conclusions

Our phylogenomic analysis (Fig 3) based on 529 nuclear genes enriched using the *Artocarpus*-Moraceae probes [89] confirms that the correct name for the Taiwanese 'breadfruit' is *Artocarpus treculianus* Elmer (Fig 4) according to the most updated taxonomic revision [86], while the true breadfruit, *A. altilis*, is extremely rare in Taiwan (Fig 2). Since *A. treculianus* is thought to be endemic to the Philippines, our study suggests that the species was likely transported from the Philippines to Lanyu in prehistorical time, providing a candidate study system to test the northward Austronesian migration into Taiwan using a commensal approach. Our study

also indicates that *Pacilo* of eastern Taiwan originated from Lanyu, while both *Pacilo* and *Cipoho* were sources of modern introductions of *A. treculianus* to other parts of Taiwan.

## Taxonomic treatment

**Artocarpus treculianus** Elmer, Leafl. Philipp. Bot. 2: 617 (1909); Merrill, Enum. Philipp. Fl. Pl. 2: 43 (1923); Jarrett, J. Arnold Arbor. 40(3): 302 (1959), *pro parte*, excl. *syn. Artocarpus nigrescens* Elmer; Gardner & Zerega, Gard. Bull. Singapore 73(2): 348 (2021). Type:—Philippine, Negros Oriental, Dumaguet, Cuenos Mountains, June 1908, *Elmer 10406* (lectotype BM, designated by Jarrett 1959 [43]; isolectotypes A, BO, L). Figs 4 & 5.

*Artocarpus ovatifolius* Merr., Philipp. J. Sci. C 9: 268 (1914). Type:—Philippines, Luzon, San Antonio, June 1912, *Ramos BS 15040* (lectotype BM, designated by Jarrett 1959 [43]; isolectotypes B, US).

*Artocarpus ovatifolius* var. *dolichostachys* Merr., Enum. Philip. Pl. 2: 43 (1923). Type:—Philippines, Samar, Apri 1914, *Ramos 1603* (lectotype BM, designated by Jarrett 1959 [43]; isolectotypes BO, GH, L, NY, P, SING).

*Artocarpus altilis* auct. non (Parkinson) Fosberg: Liu, Ill. Nat. Intr. Lign. Pl. Taiwan 2: 704 (1962); Hatusima, Mem Fac. Agric. Kagoshima Univ. 5(3): 24 (1966); Liu & Liao, Fl. Taiwan 2: 118, *pl. 233* (1976).

*Artocarpus communis* auct. non J.R.Forst. & G.Forst.: Merrill, Philipp. J. Sci., Bot. 3 (1908) 401; Sasaki, List. Pl. Formosa 151 (1928); Kudo, Iconography of Tropical Plants in Formosa 1: 6, *pl. 6* (1934); Yang, Manual of Fruit Trees in Taiwan 101, *fig. 63* (1951); Lin, Manual of Taiwan Vascular Plants 2: 46 (1999); Cheng & Lu, Botel Tabaco, Yami & Plants 48 (2000); Chung, Illustrated Flora of Taiwan 4: 298 (2017).

*Artocarpus incisus* auct. non (Thunb.) L.f.: Mackay, From Far Formosa 63 (1896); Kawakami, List Pl. Formosa. 103 (1910); Hayata, J. Coll. Sci. Imp. Univ. Tokyo 30: 278 (1911); Sasaki, J. Nat. Hist. Soc. Taiwan 3(9): 35 (1913); Chang, J. Phytogeogr. Taxon. 29(1): 2 (1981); Yamazaki, J. Phytogeogr. Taxon. 30(2): 72 (1982); Liao, Quart. J. Exp. For. Natl. Taiwan Univ. 3(1): 146 (1989); Liao, Fl. Taiwan 2$^{nd}$ 2: 137, *pl. 66* (1996).

*Artocarpus integrifolia* auct. non L.f.: Tashiro, A guide to planting trees in urban Taiwan 238 (1900); Matsumura & Hayata, J. Coll. Sci. Imp. Univ. Tokyo 22: 381 (1906).

*Artocarpus lanceolata* auct. non Trécul: Li, Woody Flora of Taiwan 115, *fig. 40* (1963).

## Supporting information

**S1 Table. Summary statistics of assembly quality of the 64 samples used in this study.**
(XLSX)

**S1 Appendix. List of literature of the Taiwanese 'breadfruit' and the misused scientific names.**
(PDF)

## Acknowledgments

The authors thank Chiayi Botanical Garden, Lanyu Township Office, National Chiayi University, National Museum of Natural Science, National Pingtung University, National Pingtung University of Science and Technology, Taipei Botanical Garden, and many local land owners for permission to sample *Artocarpus* species, Mei-Lin Chung for translating Japanese literature, Yi-Shan Chao, Hong-Wun Chen, Wei-Hsin Hu, I-Ling Lai, Shih-Hui Liu, Li-Wei Tsai, and Meng-Ying Tsai for field assistance, Chien-Wen Chen for providing important literature,

and Chang-Fu Hsieh, Szuwei Tsai, Gene-Sheng Tung, Chih-Kai Yang, and Alex Hon-Tsen Yu for valuable suggestions and discussions.

## Author Contributions

**Conceptualization:** Chi-Shan Chang, Kuo-Fang Chung.

**Data curation:** Chia-Rong Chuang, Elliot M. Gardner.

**Formal analysis:** Chia-Rong Chuang, Chia-Lun Hsieh, Elliot M. Gardner.

**Funding acquisition:** Kuo-Fang Chung.

**Investigation:** Chia-Rong Chuang, Chia-Lun Hsieh, Chi-Shan Chang, Chiu-Mei Wang, Danilo N. Tandang, Kuo-Fang Chung.

**Methodology:** Chia-Rong Chuang, Chia-Lun Hsieh, Elliot M. Gardner, Lauren Audi, Nyree J. C. Zerega.

**Visualization:** Chia-Rong Chuang, Chia-Lun Hsieh, Danilo N. Tandang, Kuo-Fang Chung.

**Writing – original draft:** Kuo-Fang Chung.

**Writing – review & editing:** Chia-Rong Chuang, Chia-Lun Hsieh, Chi-Shan Chang, Chiu-Mei Wang, Danilo N. Tandang, Elliot M. Gardner, Lauren Audi, Nyree J. C. Zerega, Kuo-Fang Chung.

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
