## [Decision Letter · Decision Letter 0]

30 Aug 2022

PONE-D-22-20826Amis Pacilo and Yami Cipoho are not the same as the Pacific breadfruit starch crop—Target enrichment phylogenomics of a long-misidentified Artocarpus species sheds light on the northward Austronesian migration from the Philippines to TaiwanPLOS ONE

Dear Dr. Chung,

Thank you for submitting your manuscript to PLOS ONE. After careful consideration, we feel that it has merit but does not fully meet PLOS ONE’s publication criteria as it currently stands. Therefore, we invite you to submit a revised version of the manuscript that addresses the points raised during the review process. We have collected two reports and as you can see from the reviewer responses the changes are very minor. Please revise your manuscript accordingly. Please particularly focus on providing the original data through Genbank and BioProject, and confirm that they are available to the public.

We look forward to receiving your revised manuscript.

Kind regards,

Christian Reepmeyer, PhD

Academic Editor

PLOS ONE

Journal Requirements:

"This study was supported by National Museum of Prehistory (No. N110011) and in parts by the Minister of Science and Technology, Taiwan (MOST 109-2621-B-001-005-MY2) and the Thematic Research Program, Academia Sinica (AS-107-TP-B18)."

6. We note that Figure 1 in your submission contain map images which may be copyrighted. All PLOS content is published under the Creative Commons Attribution License (CC BY 4.0), which means that the manuscript, images, and Supporting Information files will be freely available online, and any third party is permitted to access, download, copy, distribute, and use these materials in any way, even commercially, with proper attribution. For these reasons, we cannot publish previously copyrighted maps or satellite images created using proprietary data, such as Google software (Google Maps, Street View, and Earth). For more information, see our copyright guidelines: http://journals.plos.org/plosone/s/licenses-and-copyright.

7. We note that Figure 3 includes an image of a participant in the study. As per the PLOS ONE policy (http://journals.plos.org/plosone/s/submission-guidelines#loc-human-subjects-research) on papers that include identifying, or potentially identifying, information, the individual(s) or parent(s)/guardian(s) must be informed of the terms of the PLOS open-access (CC-BY) license and provide specific permission for publication of these details under the terms of this license. Please download the Consent Form for Publication in a PLOS Journal (http://journals.plos.org/plosone/s/file?id=8ce6/plos-consent-form-english.pdf). The signed consent form should not be submitted with the manuscript, but should be securely filed in the individual's case notes. Please amend the methods section and ethics statement of the manuscript to explicitly state that the patient/participant has provided consent for publication: “The individual in this manuscript has given written informed consent (as outlined in PLOS consent form) to publish these case details”. 

Reviewers' comments:

Reviewer's Responses to Questions

**Comments to the Author**

1. Is the manuscript technically sound, and do the data support the conclusions?

Reviewer #1: Yes

Reviewer #2: Yes

2. Has the statistical analysis been performed appropriately and rigorously? 

Reviewer #1: I Don't Know

Reviewer #2: Yes

3. Have the authors made all data underlying the findings in their manuscript fully available?

Reviewer #1: No

Reviewer #2: Yes

4. Is the manuscript presented in an intelligible fashion and written in standard English?

Reviewer #1: Yes

Reviewer #2: Yes

5. Review Comments to the Author

Reviewer #1: This paper uses new sequences from a relatively common tree taxa in Taiwan to revise its classification. It is sound research that will improve future studies through this clarification. Importantly, it also has implications for the history of Austronesians, a point that is well made in the manuscript.

I have only two minor suggestions. It might be useful to show a simplified phylogenetic tree of the taxa mentioned in the text for readers not familiar with these or their relation to one another. I would also revise the text of Figure 3 to make it clear there are unknown residents of the village in the photograph in addition to the Canadian visitor. One typo: 1,300 AD should be 1300 AD.

I searched for the underlying data on Genbank and the reported BioProject and could not find any under the identification listed. These should be confirmed as publicly available with publication of the study.

Reviewer #2: The authors perform a phylogenomic study of a Taiwanese plant taxa commonly referred to as 'breadfruit', building species trees from >500 gene trees and using the resulting phylogeny to examine multiple hypothesis concerning their taxanomic identity and their relationship with similar taxa on the neighbouring island of Lanyu. The data processing and phylogenomic methods appear to be robust and suitable for the present purpose (noting that I do not have direct experience with species tree reconstruction myself), and the authors clearly demonstrate that Taiwanese breadfruit fall within a single clade that is suggestive of a Lanyu origin, which is further attested by oral traditions of the local Indigenous peoples. Notably, the authors recognise the need for further research to help validate their claims, but overall I find the study to be well written and sufficiently informative to be deserving of publication in its current state.

I have only a few minor recommendations that are detailed below:

Line 92-94: Fig. 1. caption - please state what the different coloured labels indicate

Line 250: is this BWA aln or mem?

Line 261: 'to filtered out the column containing over 49% gaps' - should be 'to filter out the column ...'

Line 291-294: Fig. 4. caption - please state what the bolded labels indicate

6. PLOS authors have the option to publish the peer review history of their article (what does this mean?). If published, this will include your full peer review and any attached files.

Reviewer #1: No

Reviewer #2: No

---

## [Author Response · Author response to Decision Letter 0]

8 Sep 2022

Below are our responses to Reviewer 1:

Reviewer #1: This paper uses new sequences from a relatively common tree taxa in Taiwan to revise its classification. It is sound research that will improve future studies through this clarification. Importantly, it also has implications for the history of Austronesians, a point that is well made in the manuscript.

Reply: We thank Reviewer #1 for the compliment.

Reviewer #1: I have only two minor suggestions. It might be useful to show a simplified phylogenetic tree of the taxa mentioned in the text for readers not familiar with these or their relation to one another. I would also revise the text of Figure 3 to make it clear there are unknown residents of the village in the photograph in addition to the Canadian visitor. One typo: 1,300 AD should be 1300 AD.

Reply: We add a simplified phylogenetic tree of the taxa mentioned in the text in Fig. 3C. Because concern of the copyright issue, Figure 3 is removed in the revision. We also delete “,” from all years mentioned in the text (Lines 75).

Reviewer #1: I searched for the underlying data on Genbank and the reported BioProject and could not find any under the identification listed. These should be confirmed as publicly available with publication of the study.

Reply: We have deposited our data in GenBank (Bioproject no. PRJNA855987). The data is set to be release in 2026-07-05 and will be released immediately upon publication of the manuscript. For the purpose of review, please see the Reviewer link https://dataview.ncbi.nlm.nih.gov/object/PRJNA855987?reviewer=6mcs83vlkpsh7q7efe303ai465

Below are our responses to Reviewer 2: 

Reviewer #2: The authors perform a phylogenomic study of a Taiwanese plant taxa commonly referred to as 'breadfruit', building species trees from >500 gene trees and using the resulting phylogeny to examine multiple hypothesis concerning their taxanomic identity and their relationship with similar taxa on the neighbouring island of Lanyu. The data processing and phylogenomic methods appear to be robust and suitable for the present purpose (noting that I do not have direct experience with species tree reconstruction myself), and the authors clearly demonstrate that Taiwanese breadfruit fall within a single clade that is suggestive of a Lanyu origin, which is further attested by oral traditions of the local Indigenous peoples. Notably, the authors recognise the need for further research to help validate their claims, but overall I find the study to be well written and sufficiently informative to be deserving of publication in its current state.

Reply: We thank Reviewer #2 for the compliment.

Reviewer #2: Line 92-94: Fig. 1. caption - please state what the different coloured labels indicate.

Reply: We thank Reviewer #2 for the suggestions. Captions of Fig 1 (Line 93–97) and 3 (Line 293–297) have been modified to explain the meaning of colors used in both figures.

Reviewer #2: is this BWA aln or mem?

Reply: We thank Reviewer #2 for the questions. In the script of HybPiper pipeline [which are available on the HybPiper GitHub site (https://github.com/mossmatters/HybPiper)], it states that it is BWA mem used. As HybPiper is cited and freely available, we don’t think it is necessary to specify this in the manuscript.

Reviewer #2: Line 261: 'to filtered out the column containing over 49% gaps' - should be 'to filter out the column ...'

Reply: We thank Reviewer #2 to spot this typo. We have corrected it (Line 262).

Reviewer #2: Line 291-294: Fig. 4. caption - please state what the bolded labels indicate.

Reply: We thank Reviewer #2 for the comments. Captions of Fig 1 (Line 93–97) and 3 (Line 293–297) have been modified to explain the meaning of colors and bold-faced used in both figures.

---

## [Editor Report · Decision Letter 1]

12 Sep 2022

Amis Pacilo and Yami Cipoho are not the same as the Pacific breadfruit starch crop—Target enrichment phylogenomics of a long-misidentified Artocarpus species sheds light on the northward Austronesian migration from the Philippines to Taiwan

PONE-D-22-20826R1

Dear Dr. Chung,

We’re pleased to inform you that your manuscript has been judged scientifically suitable for publication and will be formally accepted for publication once it meets all outstanding technical requirements.

Kind regards,

Christian Reepmeyer, PhD

Academic Editor

PLOS ONE
---

## [Editor Report · Acceptance letter]

22 Sep 2022

PONE-D-22-20826R1 

Amis Pacilo and Yami Cipoho are not the same as the Pacific breadfruit starch crop—Target enrichment phylogenomics of a long-misidentified Artocarpus species sheds light on the northward Austronesian migration from the Philippines to Taiwan 

Dear Dr. Chung:

I'm pleased to inform you that your manuscript has been deemed suitable for publication in PLOS ONE. Congratulations! Your manuscript is now with our production department. 

Kind regards, 

on behalf of

Dr. Christian Reepmeyer 

Academic Editor

PLOS ONE